# Electrochemical desulfurative borylation of thiols, disulfides, thioethers and thioacetals

Julius Kuzmin [1], Cristiana Margarita [1,2,3], Johannes Winter[1,3] & Helena Lundberg [1] ✉

Low-valent sulfur-containing compounds are abundant among natural and synthetic products but remain underutilized as starting materials in desulfurative transformations. Herein, we present thiols, disulfides, thioethers, and thioacetals as precursors in a direct desulfurative electrochemical process for the formation of alkylboronic esters, including late-stage functionalization of pharmaceutically relevant scaffolds and natural products. The electrochemical protocol is simple, user-friendly and scalable, successfully producing gram quantities of borylated product.

Alkylboronic acids and esters are of great importance in chemical science. In the context of medicinal chemistry, the introduction of boronic acids can improve activity and/or pharmacokinetic properties[1–5], whereas the reversible dynamic covalent bonds of boronic esters have enabled the synthesis of self-healing macromolecular materials[6]. Furthermore, the compound classes are widely used in organic synthesis as reagents and coupling partners. For example, allylic boronic acids and esters are well established as reagents for allylboration of aldehydes[7], Prins cyclization[8], and Suzuki-Miyaura coupling[9]. In the context of benzylic boronic species, pinacol esters have been used as alkyl donors for Pd-catalyzed cross-couplings[10–13], Cu-catalyzed C-O and C-N couplings of Chan-Lam-Evans type[10,14], Minisci-type reactions[15,16], and various C-C bond formations[17–19], as well as starting materials for oxidative conversion into alcohols and ketones (Fig. 1A)[10,20]. Furthermore, the related class of benzylic tetrafluoroborates have been used in a variety of bond-forming reactions under electrochemical conditions[21]. There are a variety of synthetic routes to organoboron compounds[22], including hydroboration of olefins[23–25] and addition of organometallic nucleophiles to boron electrophiles[26]. Furthermore, borylative cross-couplings of alkyl and aryl (pseudo)halides using transition metal catalysis is a common strategy[27–36], as well as borylation via C-H activation by means of photoredox catalysis[37–42], and photoelectrocatalysis[42,43], using diboron species as coupling partners. Borylative electrochemical routes with such diboron species have also been disclosed starting from aryl and alkyl halides[44–50], redox active N-hydroxyphthalimide esters[51,52] and amine derivaties[53–56]. In a few instances, boranes were used as cross-coupling partners for borylation of alkyl halides with Ti-catalysis, and for deoxygenative borylation of alcohols and their oxidized analogs under electrochemical conditions[57,58]. However, thiols and thiol derivatives remain scarce as starting materials in borylative transformations, despite their abundance in natural and synthetic products from the agrochemical, fragrance, and pharmaceutical industries[59–62]. While oxidized or charged species such as sulfoxides, sulfones, and sulfonium salts are used as starting materials in transition metal catalyzed protocols[22,63], neutral, low-valent analogs are easily counted. A few reports on desulfurative borylation of thioethers have been disclosed, displaying full selectivity for borylation of aryl side-chains (Fig. 1B)[64,65], analogous to that reported under photochemical oxidative conditions[66]. In addition, photoredox catalysis has been used to form alkylboronic acids and esters from cysteine or cysteine-derived thioethers using super-stoichiometric amounts of phosphine reagents (Fig. 1C)[67,68]. Encouraged by our previous successful use of benzylic thiol derivatives as alkyl donors under electroreductive conditions[69], we set out to explore desulfurative borylation using this class of compounds under electrochemical conditions (Fig. 1D).

## Results

### Optimization of reaction conditions

Inspired by the work of Lin and co-workers[57], we set out to explore the use of pinacol borane (HBpin) as coupling partner for the targeted desulfurative borylation of model compound benzyl phenyl sulfide (1a). Evaluation of the reaction parameters (Table 1) revealed that inexpensive graphite electrodes in an undivided cell with three equivalents of HBpin in THF with tetrabutylammonium borohydride

[1]Department of Chemistry, KTH Royal Institute of Technology, Stockholm, Sweden. [2]Present address: Department of Basic and Applied Sciences for Engineering (SBAI), Sapienza University of Rome, Rome, Italy. [3]These authors contributed equally: Cristiana Margarita, Johannes Winter.
✉e-mail: hellundb@kth.se

**A. Benzyl boronic esters as gateway to multiple product categories**

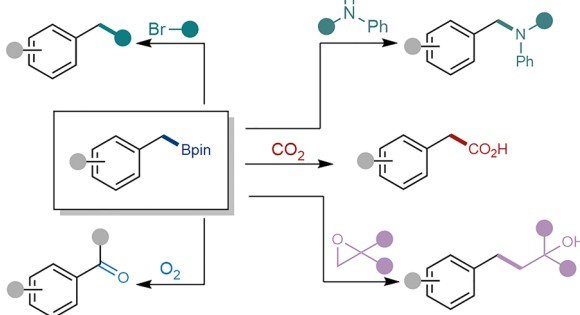

**B. Transition metal-catalyzed borylation of thioethers (Ref. 64, 65)**

**C. Photochemical desulfurative borylation (Ref. 67, 68)**

● = H, Ar

**D. Electrochemical borylation of thiols and thiol derivatives (this work)**

● = H, SR, Ph

• Direct borylation • Mild conditions • High yields • Scalable •

**Fig. 1 | Utilization and synthesis of boronic esters. A** Transformation of benzyl-boronic esters to various product categories. **B** Transition metal-catalyzed borylation of thioethers. **C** Photochemically driven desulfurative borylation. **D** Electrochemically driven borylation of thiols and thiol derivatives.

**Table 1 | Evaluation of reaction conditions[a]**

$C_{gr}$ | $C_{gr}$
10 mA, 2 F

$NBu_4BH_4$ (0.1 M)
THF, r.t., $N_2$
HBpin (3 equiv.)

1a → 2a

| Entry | Deviation from above | 2a (%)[b] |
|---|---|---|
| 1 | – | 99 |
| 2 | $B_2cat_2$ (3 equiv.)[c] | 8 |
| 3 | MeCN | 33 |
| 4 | MeCN with 10 equiv. HBpin | 78 |
| 5 | Non-anhydrous | 0 |
| 6 | Under air | 60 |
| 7 | 1.5 equiv. HBpin | 99 |
| 8 | No electricity (24 h) | 0 |

[a]Graphite electrodes, constant current 10 mA, 1a (0.5 mmol), HBpin (3 equiv.), $NBu_4BH_4$ (0.5 mmol), THF (5 mL), r.t., $N_2$, 2 F, undivided cell. For complete set of optimization data, see Supplementary Information, Section S1.
[b]Yield determined by high-performance liquid chromatography (HPLC).
[c]Transesterification with pinacol was carried out subsequent to electrolysis.

($NBu_4BH_4$) as supporting electrolyte furnished the boronic ester (2a) in quantitative yield after passing 2 F at 10 mA under inert and anhydrous conditions (Table 1, entry 1). A switch from HBpin to bis(catecholato) diboron ($B_2cat_2$) resulted in considerably lower yields of 2a subsequent to transesterification (Table 1, entry 2). The use of acetonitrile (MeCN) as solvent resulted in a lower yield of 2a even with increased amounts of HBpin (Table 1, entries 3 and 4). Non-anhydrous conditions failed to furnish 2a (Table 1, entry 5) and exchanging the $N_2$ atmosphere for air resulted in a lower yield (Table 1, entry 6), highlighting the importance of dry and inert conditions. Furthermore, it was found that the amount of HBpin could be reduced from 3 to 1.5 equivalents with maintained product yield (Table 1, entry 7) and these conditions were henceforth used. No reaction occurred in the absence of electricity (Table 1, entry 8), confirming that the transformation is electrochemically triggered.

**Substrate scope**

With optimized conditions at hand, the generality of the borylation protocol was assessed (Fig. 2A). Gratifyingly, a variety of benzylic thioethers were successfully transformed into the corresponding pinacolboronic esters in good yields, including substrates with reductively labile functionalities such as aryl halides (2b-d) and a benzylic trifluoromethyl group (2e). Furthermore, substituents such as Bpin (2g), methoxy groups (2h and 2i), acetal (2j), and methyl thioether (2k) were tolerated, the latter with full selectivity for benzylic C-S bond cleavage. Similarly, a methyl ester was tolerated under the reaction conditions, forming product 2f in a moderate yield. Moreover, the protocol is compatible with various heterocycles, including furan (2l),

thiophene (2m), morpholine (2n), and N-methyl indole (2o). The alkene-containing boronic ester 2p formed in moderate yield, and the di-borylated product (2q) was smoothly synthesized from the corresponding di-thioether upon the use of twice the amount of pinacolborane and 2.5 times the charge.

Furthermore, thioethers with multiple substituents on the phenyl ring formed the target boronic esters in moderate yields (2v and 2w). The tertiary and secondary pinacolboronic esters 2r-2u were successfully formed under standard conditions, with generally higher yields for the less sterically hindered starting materials. Interestingly, preferential borylation of a tertiary over a secondary thioether was observed for the formation of product 2y from a derivative of the antihistamine terfenadine, indicating that the selectivity may rely on a combination of steric and electronic effects. The phenyl thioether based on the muscarinic antagonist diphenidol formed the corresponding borate ester 2x in moderate yield, whereas the phenyl thioethers based on the core structure of rosuvastatin, used for treatment of cardiovascular disease, the phosphodiesterase inhibitor roflumilast, and acetylated salicin formed the targeted borylation products in good yields (2z, 2aa, and 2ab, respectively). Furthermore, a selection of thioacetals was assessed under slightly modified standard conditions (Fig. 2B). Gratifyingly, these conditions successfully furnished the corresponding mono-borylated products in fair yields (2a, 2ac-2ae), including a derivative of fenofibrate, a lipid-modifying agent used in the treatment of hypertriglyceridemia and mixed dyslipidemia. Due to the use of thioacetals as protecting groups of aldehydes and ketones as well as starting materials for C-C bond formation in the Corey-Seebach reaction[70,71], this borylation strategy may find its use in coupled sequences for the synthesis of complex organic compounds ahead. In addition, allylic thioethers were amenable to desulfurative borylation (Fig. 2C), furnishing allylic boronic esters derived from myrtenol, geraniol, and perillyl alcohol in good yields (2af-2ah). Here, the formation of regioisomers was observed with clear preference for primary, less sterically hindered sites. Notably, the borylation procedure proved scalable, furnishing 1 gram of product 2a from 1a, demonstrating the utility of the procedure for practical larger batch preparations (Fig. 2D). Finally, a selection of compounds with differently substituted aryl groups were assessed to probe the protocol's tolerance towards the aryl sidechain of the thioether (1a-1f). Interestingly, lower yields of the benchmark product 2a were obtained in all

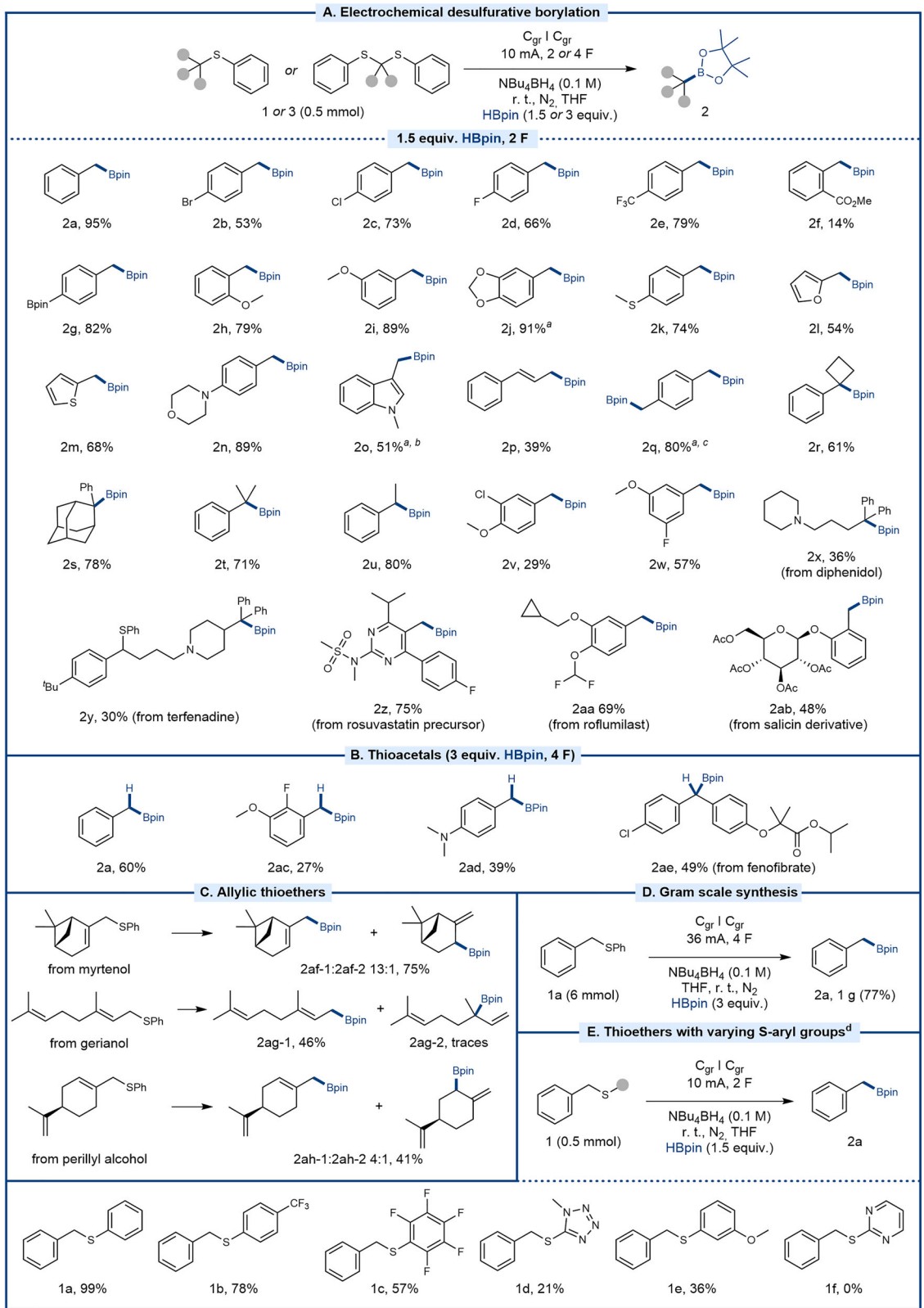

**Fig. 2 | Electrochemical desulfurative borylation of aryl thioethers and thioacetals. A** Borylation of thioethers. **B** Borylation of thioacetals. **C** Borylation of allylic thioethers. **D** Synthesis on gram scale. **E** Effect of arene substitution on thioether borylation. All yields refer to isolated yields unless otherwise noted. [a] 3 equiv. HBpin [b] 4 F [c] 5 F [d] HPLC yield.

cases (Fig. 2E), demonstrating the unsubstituted phenyl group to be preferential.

To probe whether the S-aryl group was crucial for borylation to occur, two symmetric disulfides were assessed as starting materials

(Fig. 3A), gratifyingly furnishing the targeted products 2a and 2l in 77% and 35% yield. By monitoring the reaction starting from disulfide 4a over time via sampling and off-line HPLC analysis, it was found that the transformation proceeds via initial reductive cleavage of 4a to form the

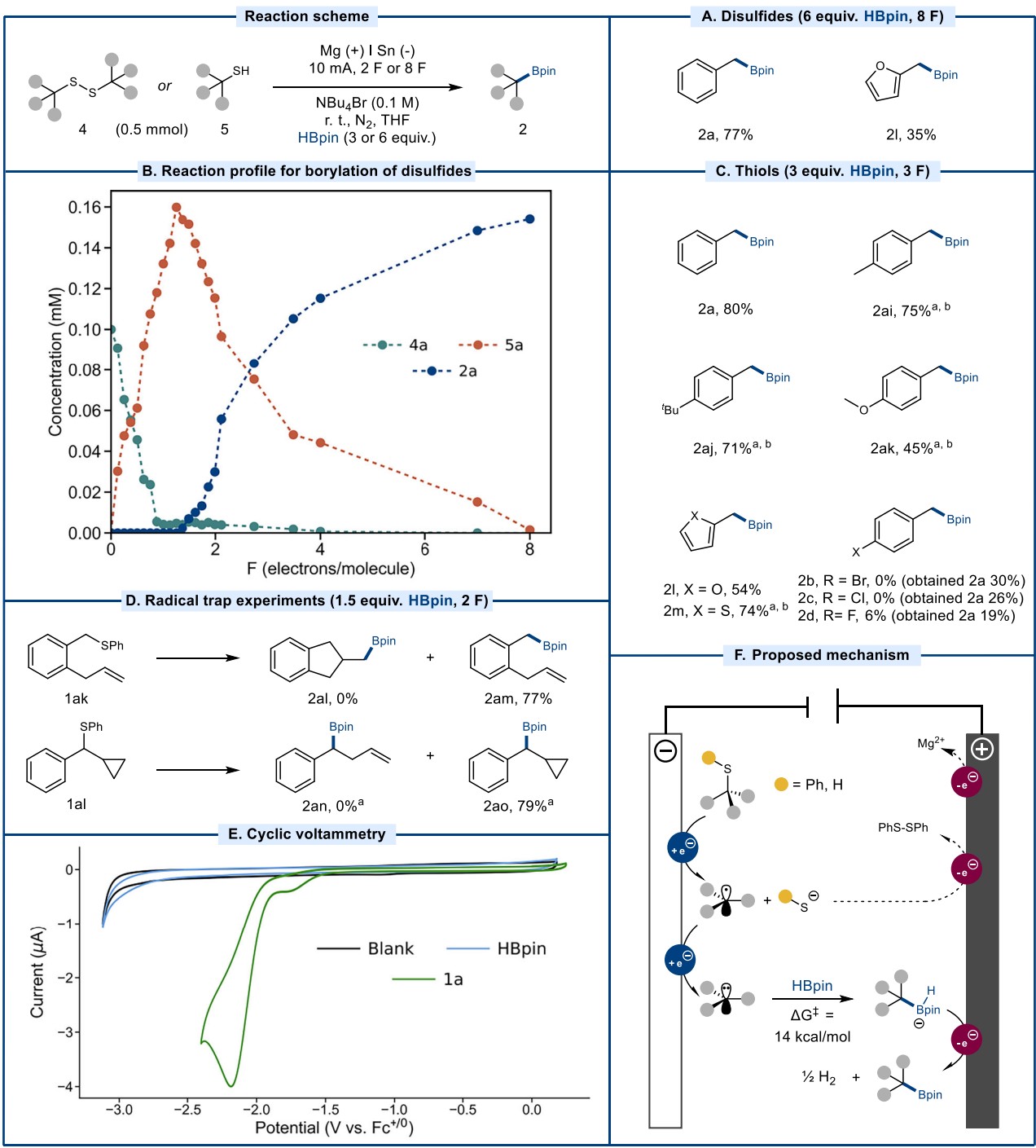

**Fig. 3 | Desulfurative borylation. A** Borylation of disulfides. **B** Reaction profile for electrolytic borylation of 4a. **C** Borylation of thiols. **D** Radical trap experiments. **E** Cyclic voltammetry study of compound 1a and HBpin. **F** Proposed mechanism for borylation of thioethers and thiols. Yields are isolated unless otherwise noted. [a]qNMR yield. [b]4F.

corresponding thiol 5a that, in turn, is transformed into the borylated product 2a (Fig. 3B). To probe whether such direct -SH bond cleavage could present synthetic benefits, a small selection of thiols was assessed. Gratifyingly, a significant increase in yield was obtained for 2l using thiol as starting material instead of the corresponding disulfide (Fig. 3C). Furthermore, a selection of *p*-substituted thiols resulted in the corresponding boronic ester products in good yields (2ai-2ak), as did the 2-thiophene thiol (2m). However, halide-substituted thiols were found to undergo reductive dehalogenation in addition to desulfurative borylation, in contrast to the reactivity of their thioether counterparts that maintained their halides throughout the reaction

(Fig. 1A). This reactivity difference clearly demonstrates that the S-phenyl sidechain has a beneficial effect on selectivity, a feature that may be understood in light of the anodic shift in reduction potential that it induces in the starting material[69,72]. This phenyl sidechain can be be easily installed onto the thiol using a variety of synthetic procedures[73–75], including transition metal catalysis[76–100].

## Mechanistic studies

To probe the mechanism of the borylation reaction, a radical trap experiment was carried out using a thioether equipped with an alkene sidechain (1ak). When subjected to standard conditions, 1ak did not

convert to the 5-exo-trig product 2al (Fig. 3D). Instead, the linear product 2am was obtained in 77%, indicating that the transformation does not proceed via a C-centered radical with a lifetime sufficient for cyclization. Analogously, thioether 1al did not undergo borylative ring-opening but formed the cyclopropyl-containing borate ester 2ao in 79% yield, further supporting the hypothesis of short lifetimes for intermediate benzylic radicals. These combined results are in line with our previous findings[69], in which benzylic phenyl thioethers were demonstrated to undergo reductive C-S bond cleavage to form carbanions via a rapid radical-polar crossover under electroreductive conditions, with the aryl mercaptan sidechain eventually forming the corresponding disulfide upon anodic oxidation. Furthermore, the formation of carbanionic intermediates is consistent with the observed mixture of regioisomers for allylic substrates, resulting from carbanion delocalization. This delocalization favors positioning of the carbanion in primary position, effectively resulting in borylation at the sterically more accessible site (Fig. 2C). Based on these findings, we propose that the desulfurative borylation proceeds via a (semi-)paired electrolytic mechanism (Fig. 3F). Starting from thiols or thioethers, an initial cathodic SET sets off a mesolytic C-S bond cleavage to furnish a benzylic open-shell species. This intermediate rapidly undergoes a reductive radical-polar crossover via a second SET to furnish a carbanion, which reacts with the reductively stable pinacolborane electrophile to form an anionic borohydride intermediate, with an associated energy barrier of around 14 kcal/mol for the C-B coupling step as determined by DFT calculations. Finally, we propose that the resulting borohydride intermediate undergoes subsequent oxidation at the anode to furnish the neutral boronic ester product along with hydrogen gas. Such electrochemical oxidation of borohydrides is in line with our previous studies[101], although alternative chemical routes to the formation of the neutral borylated product cannot be ruled out[57,102–104].

## Conclusions

In this work, an efficient desulfurative electrochemical protocol for the synthesis of alkylboronic esters from thiols, disulfides, thioacetals, and thioethers is presented. Mechanistically, the borylation proceeds via carbanionic intermediates with pinacolborane as unconventional electrophilic coupling partner. The (semi-)paired electrolytic transformation tolerates a wide range of functional groups and was successfully applied to pharmaceutically relevant scaffolds and natural products. Complete selectivity for activation of the C(sp$^3$)-S bond in aryl alkyl thioethers and acetals was observed, orthogonal to that of transition metal catalyzed protocols. With its operational simplicity and scalable nature, this electrochemical method presents an attractive avenue to synthetically useful alkylboronic esters.

## Methods

### General procedure for borylation of thioethers and thioacetals

To an oven-dried 10 mL-ElectraSyn vial equipped with a magnetic stir bar (dimensions 15 mm × 6 mm), graphite electrodes, the starting material (1.0 equiv., 0.50 mmol), and NBu$_4$BH$_4$ (1.0 equiv., 0.50 mmol, 130 mg) were added. The mixture was evacuated and back-flushed with nitrogen three times before anhydrous stabilizer-free THF (5 mL) was added, followed by HBpin (0.75 mmol for thioethers and 1.5 mmol for thioacetals). The reaction was carried out by applying 10 mA (~10 mA/cm$^2$) at room temperature for the indicated charge (2 F for thioethers and 4 F for thioacetals) with a stir rate of 750 rpm. After electrolysis, the solvent was removed under reduced pressure and the crude reaction mixture was dissolved in EtOAc, washed with an aqueous solution of NH$_4$Cl (30 mL), and extracted with EtOAc (15 mL × 3). The combined organic phases were dried over sodium sulfate and purified by column chromatography on oven-dried silica gel to provide the desired product.

### General procedure for borylation of disulfides and thiols

To an oven-dried 10 mL-ElectraSyn vial equipped with a magnetic stir bar (dimensions 15 mm×6 mm), a magnesium anode and a tin cathode, the starting material (1.0 equiv., 0.50 mmol) and NBu$_4$Br (1.0 equiv., 0.50 mmol, 161 mg) were added. The mixture was evacuated and back-flushed with nitrogen three times before anhydrous stabilizer-free THF (5 mL) was added, followed by HBpin (3.0 mmol for disulfides and 1.5 mmol for thiols). The reaction was carried out by applying 10 mA (~10 mA/cm$^2$) at room temperature for the indicated charge (8 F for disulfides and 3 F for thiols) with a stir rate of 750 rpm. After electrolysis, the solvent was removed under reduced pressure and the crude reaction mixture was dissolved in EtOAc, washed with an aqueous solution of NH$_4$Cl (30 mL), and extracted with EtOAc (15 mL x 3). The combined organic phases were dried over sodium sulfate and purified by column chromatography on oven-dried silica gel to provide the desired product.

## Data availability

The data generated in this paper are provided within the article and its Supplementary Information file. Data supporting the findings of this manuscript are also available from the corresponding author upon request.

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

## Acknowledgements

We thank Dr. Guillermo Ahumada (G. A.) and M.Sc. Nils Schwarz (N. S.) for fruitful discussions, and N. S. for assistance with the synthesis of the thioether precursors for products 2 h, 2j, 2 m, 2n, and 2r. Financial support to the corresponding author from the Swedish Research Council (grant no. 2021-05551), the European Research Council (grant no. 101164660), the Wenner-Gren foundation, Magnus Bergvalls stiftelse, Stiftelsen Lars Hiertas Minne, and KTH Royal Institute of Technology is gratefully acknowledged.

## Author contributions

Julius Kuzmin (J.K.): Conceptualization, Data curation, Formal analysis, Investigation, Methodology, Project administration, Validation, Visualization, Writing—original draft, Writing—review & editing. Cristiana Margarita (C.M.): Data curation, Formal analysis, Investigation, Methodology, Validation, Visualization, Writing—review & editing. Johannes Winter (J.W.): Data curation, Formal analysis, Investigation, Methodology, Validation, Visualization, Writing—review & editing. Helena Lundberg (H.L.): Conceptualization, Data curation, Formal analysis, Project administration, Visualization, Writing—original draft, Writing—review & editing, Funding acquisition, Resources, Supervision

## Funding

## Competing interests

The authors declare no competing interests.
