## [Transparent Peer Review file · Nature Communications]

Electrochemical Desulfurative Borylation of Thiols, Disulfides, Thioethers and Thioacetals

Corresponding Author: Professor Helena Lundberg

Version 0:

Reviewer comments:

Reviewer #1

(Remarks to the Author)

In this article, the authors have achieved the direct desulfurization using disulfides, thioethers and thiols as precursors to form alkylboronic esters in an electrochemical process, including late-stage functionalization of pharmaceutically relevant frameworks and natural products. The (semi-)paired electrolytic transformation proceeds with high Faradaic efficiency, tolerates various functional groups. Remarkably, the transformation proceeds with complete selectivity for activation of the C(sp³)-S bond in aryl alkyl thioethers, orthogonal to that of transition metal catalyzed protocols. This electrochemical method presents an attractive avenue to alkylboronic esters via carbanionic intermediates and pinacolborane. However, there are quite a lot of issues in the SI (Supplementary Information) section that need to be carefully revised. This article is currently not suitable for publication in Nature Communications.

1. The existence of the anionic borohydride intermediate and its oxidation to the final borylation product should be further verified, including the isolation and CV analysis, DFT calculations etc.
2. For substrate 2f, the authors claimed to achieve the product with a good yield. However, only a 14% yield was obtained. Have the authors ever tried substrates with two substituents on the benzene ring?
3. For substrate 2f, have the authors tried using electron-donating groups? Could substrates with electron-donating groups achieve higher yields?
4. Non-graphite electrodes were used for the thiol compounds in Figure D. Have the authors tried graphite electrodes? If so, would the dehalogenation product also be obtained?
5. The first "Figure E" in the text should be corrected to "Figure D."
6. The description in the text, "Furthermore, the formation intermediates is consistent with the mixture of regioisomers that was obtained from allylic thioethers (Scheme 1C), likely the result of intermediate allylic anion," is unclear. Please provide a clearer explanation.
7. In the text, the statement, "Furthermore, the two electron equivalents that theoretically would be required for the formation of a carbanionic intermediate correspond well with the experimental conditions, under which full conversion of thioether starting materials is achieved after 2F with up to 95% isolated yield of the borylated product," is not entirely accurate. Such conditions cannot be applied to substrates like thiols and disulfides.
8. In the SI, please re-purify the spectra of product 2ab and re-check the boron spectra of 2n and 2q. The carbon spectra of 1s and 1l have insufficient resolution; please include the original spectra. The boron spectrum of 2k has a similar issue. The hydrogen spectrum of 1n lacks corresponding integration values, and the hydrogen spectrum of 1o requires purification.
9. There are numerous issues in the SI, including inaccurate hydrogen spectral data, inconsistent font styles, and discrepancies in the number of decimal places retained for carbon spectra. In addition, the integrations of the NMR spectra are not precise. Please review and correct these issues carefully.
10. In the section "General procedure for borylation of disulfides and thiols," the format of "NBu₄Br, (~10 mA/cm²), NH₄Cl" is incorrect. Please revise it.

Reviewer #2

(Remarks to the Author)

Lundberg and colleagues synthesized alkylboronic esters through the electrochemical desulfurization of thiols, disulfides,

and thioethers. The reaction involves two SET processes, where electroreduction generates a carbanion, which is subsequently captured by electrophilic boron reagents to form the C-B bond. While this method is simple and interesting, this reviewer considers that the work, in its current form, is somewhat preliminary and may not yet be suitable for publication in Nature Communications.

The main issue with this work is its novelty. Previous studies have successfully achieved the synthesis of alkylboronic esters through photocatalysis (Nature 2020, 586, 714-719; Science 2024, 383, 537-544), electrocatalysis, and photoelectrocatalysis (Nat. Synth. 2024, 3, 537-544; Nat. Commun. 2023, 14, 6530). Notably, these methods demonstrate a broad substrate scope and yield favorable results. At the same time, photocatalysis has also been successfully employed to achieve desulfurization into alkylboronic esters (Org. Lett. 2021, 23, 10, 3919-3922). However, these references are not cited in the manuscript. Compared to previous works, the substrates in this study are quite limited, and the compatibility of substrates is also poor. The reaction in this study only includes activated benzyl and allyl sulfides—what about alkyl sulfides and aryl sulfides? Furthermore, in the section on substrate expansion, the authors did not demonstrate the effect of substituents on the reaction of sulfur-containing phenyl groups, which should be addressed to better explain the influence of substituents. The authors emphasize the use of alkylboronic esters in the introduction but do not conduct subsequent derivative reactions to verify the practical utility of the products.

Since the authors have verified that thiol is an intermediate in the reaction, and it can also directly react with relatively high yield, why not directly use thiols as a substrate for reaction screening? Thiol substrates are cheaper and more readily available, which would make the study more meaningful, and would be a more significant contribution to the construction of alkyl or arylboronic esters.

Moreover, the paper offers little expansion on the thiols and disulfides substrates, and does not reflect the compatibility of these two types of substrates. In the radical trapping experiment, the lifetime of the primary free radical is short and is easily quenched, making it difficult to obtain cyclization products. More stable tertiary radicals should be considered for this experiment. The author cited literature [51] in the article, but it is not listed in the references.

Reviewer #3

(Remarks to the Author)

Version 1:

Reviewer comments:

Reviewer #1

(Remarks to the Author)

I already reviewed this manuscript. After a careful look into the revised version of the manuscript, the answers, and the added data from the authors, I am fully happy with this novel draft. This work is now suitable for publication in Nat Commun. Hence, I recommend this manuscript for publication in its current form.

Reviewer #2

(Remarks to the Author)

Although the authors have addressed most of the requested revisions and responded clearly to the reviewers' comments, certain limitations remain. Specifically, the substrate scope is still rather narrow, as only activated benzyl and allyl sulfides are tolerated in the reaction. While both HBpin and diborane compounds are inexpensive, we do not believe its use offers a significant advantage. Moreover, compared to thiols and their derivatives, which are often toxic and malodorous, the halides such as benzyl bromides are generally more favorable as substrates. In fact, most of the sulfide substrates used in Scheme 1 were prepared from benzyl bromides and thiophenols, which further reduces the novelty of the method. For these reasons, we considered that the revised manuscript does not meet the publication standards of Nature Communications.

Reviewer #3

(Remarks to the Author)

Lundberg and co-workers have addressed most of the reviewers' concerns and the quality of the manuscript has also improved significantly. However, I still have serious concerns about the significance of this electrochemical method, because benzyl bromides can be directly converted into alkyl boronic esters by electrochemical synthesis. Converting benzyl bromides into benzylic thioethers first and then into benzylic boronic esters by electrochemical method seems like an unnecessary step. To highlight the significance of this method, natural thioethers should be used as the substrate as much as possible. However, there are still very few relevant examples. Based on these concerns, the reviewer could not support the publication of this manuscript in Nature Communications.

Reviewer #1 (Remarks to the Author):

1. The existence of the anionic borohydride intermediate and its oxidation to the final borylation product should be further verified, including the isolation and CV analysis, DFT calculations etc.

Reply: The proposed borohydride intermediate is previously described in the literature (references 57, and 101-103) but has, to the best of our knowledge, never been isolated due to its inherent lability. Nevertheless, our new DFT calculations (see the updated Supplementary Information, Section S8) do support the proposed mechanism, indicating that the barrier for the carbanionic attack onto the borane for the formation of the borohydride intermediate is around 14 kcal/mol. While chemical routes to the formation of the neutral borylated product cannot be ruled out, the hypothesized rapid anodic oxidation of the borohydride intermediate is supported by our recent work (reference 100).

2. For substrate 2f, the authors claimed to achieve the product with a good yield. However, only a 14% yield was obtained. Have the authors ever tried substrates with two substituents on the benzene ring?

Reply: We thank the reviewer for pointing this out and the text has been modified to reflect that **2f** was obtained in a low yield. A selection of disubstituted aryl thioethers and one thioacetal were tested and the successfully formed products **2v**, **2w** and **2ac** are now found in the revised Scheme 1.

3. For substrate 2f, have the authors tried using electron-donating groups? Could substrates with electron-donating groups achieve higher yields?

Reply: An electron-rich analogue of **2f** – compound **2h** – could be formed in high yield (79%) and has been added to the revised Scheme 1. This yield is on par with other compounds bearing electron-donating groups, e.g., **2i-k**.

4. Non-graphite electrodes were used for the thiol compounds in Figure D. Have the authors tried graphite electrodes? If so, would the dehalogenation product also be obtained?

Reply: We thank the reviewer for this comment and have carried out a few additional experiments to probe the behaviour of the thiol reduction using graphite electrodes in the presence of borohydride. As can be seen in the revised Supplementary Information (Section S1.6), the different conditions result in similar product outcomes with the dehalogenated product being observed in both cases. However, the electrode material can indeed influence the reaction outcome, as evident from the borylative electrolysis of disulfide **4a**, furnishing product **2a** in 30% yield using graphite electrodes in the presence of NBu_4BH_4 and in 77% yield using a Mg anode and Sn cathode (Table S5, entries 1-2 in Supplementary Information).

5. The first "Figure E" in the text should be corrected to "Figure D."

Reply: All figures have been updated in the revised manuscript.

6. The description in the text, "Furthermore, the formation intermediates is consistent with the mixture of regioisomers that was obtained from allylic thioethers (Scheme 1C), likely the result of intermediate allylic anion," is unclear. Please provide a clearer explanation.

Reply: We thank the Reviewer for pointing out this confusing sentence and have revised the text for clarity.

7. In the text, the statement, "Furthermore, the two electron equivalents that theoretically would be required for the formation of a carbanionic intermediate correspond well with the experimental conditions, under which full conversion of thioether starting materials is achieved after 2F with up to 95% isolated yield of the borylated product," is not entirely accurate. Such conditions cannot be applied to substrates like thiols and disulfides.

Reply: We acknowledge the misleading wording and have removed the sentence from the text.

8. In the SI, please re-purify the spectra of product 2ab and re-check the boron spectra of 2n and 2q. The carbon spectra of 1s and 1l have insufficient resolution; please include the original spectra. The boron spectrum of 2k has a similar issue. The hydrogen spectrum of 1n lacks corresponding integration values, and the hydrogen spectrum of 1o requires purification.

Reply: We thank the reviewer for this comment. Spectrums for compound **1s** **1l** and **2k** (now **1y**, **1g**, and **2l**) and has been fixed, The products formerly labelled **2ab** and **2q** (now **2ah** and **2r**) are isolated as an inseparable mixture with the HBpin-tributylamine complex, which is previously reported in the literature (reference 23 in Supplementary Information). This borane-amine complex gives rise to the additional signal in ¹¹B-NMR. For the compound formerly labelled as **2n** (now **2o**), the compound was isolated as an inseparable mixture with pinacolboronic acid that gives rise to the additional signal in ¹¹B-NMR (a reference value can be found in reference 20 in Supplementary Information). The matter has been clarified in the revised Supplementary Information, and the yields have been recalculated. The compound formerly denoted **1o** (now **1u**) has been repurified and the associated spectra have been updated in the revised Supplementary Info.

9. There are numerous issues in the SI, including inaccurate hydrogen spectral data, inconsistent font styles, and discrepancies in the number of decimal places retained for carbon spectra. In addition, the integrations of the NMR spectra are not precise. Please review and correct these issues carefully.

Reply: We thank the reviewer for this comment and have updated the Supplementary Information accordingly.

10. In the section "General procedure for borylation of disulfides and thiols," the format of "NBu4Br, (10 mA/cm²), NH₄Cl" is incorrect. Please revise it.

Reply: These typos have been corrected.

Reviewer #2 (Remarks to the Author)

1. *The main issue with this work is its novelty. Previous studies have successfully achieved the synthesis of alkylboronic esters through photocatalysis (Nature 2020, 586, 714-719; Science 2024, 383, 537-544), electrocatalysis, and photoelectrocatalysis (Nat. Synth. 2024, 3, 537-544; Nat. Commun. 2023, 14, 6530). Notably, these methods demonstrate a broad substrate scope and yield favorable results. At the same time, photocatalysis has also been successfully employed to achieve desulfurization into alkylboronic esters (Org. Lett. 2021, 23, 10, 3919-3922). However, these references are not cited in the manuscript.*

Reply: We thank the reviewer for pointing out these references that we have added to the manuscript (references 37, 38, and 43), along with a discussion of these described strategies. For the novelty, this work uses thiols and their derivatives to furnish borylated products under electrochemical conditions, which, to the best of our knowledge, has not previously been reported. In addition to an expanded scope of these compound classes, the revised manuscript now also contains examples of the successful conversion of thioacetals into monoborylated products – a compound class that, to the best of our knowledge, has previously not been used as alkyl donor in borylation reactions under any set of synthetic conditions. Furthermore, this work describes the use of HBpin as coupling partner in borylation, rather than the classic diboron reagents that the majority of borylation protocols rely on (e.g., the new references 37, 38, and 43). The use of HBpin as electrophile is highly unconventional, the borylation proceeds via a different mechanism compared to the established radical routes with diboron reagents and remains scarce in the chemical literature. For these reasons, we do not agree with the view on this work's novelty expressed by the Reviewer.

2. *Compared to previous works, the substrates in this study are quite limited, and the compatibility of substrates is also poor. The reaction in this study only includes activated benzyl and allyl sulfides—what about alkyl sulfides and aryl sulfides?*

Reply: We thank the reviewer for the feedback and have assessed a selection of alkyl and aryl sulfides for the reductive borylation. Unfortunately, alkyl sulfides do not react under this set of conditions (see Supplementary Information, Section S7). Similarly, diphenyl sulfide reacts poorly under the applied conditions and resulted in 0% yield of the borylated product using general procedure 1 and 4% yield using general procedure 2 for 8 F with 6 equiv. HBpin. This reactivity is in line with what may be anticipated for a carbanionic mechanism. The information has been added to the Supplementary Information (Section S6).

3. *Furthermore, in the section on substrate expansion, the authors did not demonstrate the effect of substituents on the reaction of sulfur-containing phenyl groups, which should be addressed to better explain the influence of substituents.*

Reply: We thank the reviewer for this input and have added the results using a selection of S-aryl groups in Scheme 1 in the main text. Interestingly, these alternative aryl substituents all resulted in lower yields of the benchmark product, seemingly due to side-reactions.

4. *The authors emphasize the use of alkylboronic esters in the introduction but do not conduct subsequent derivative reactions to verify the practical utility of the products.*

Reply: We acknowledge the point raised by the reviewer and note that there is already a significant number of methods available that demonstrate such practical utility for the formed product class. To highlight this better, the key references 7-20 have been added to the revised introduction, along with a new section in Figure 1 that showcases selected examples.

5. *Since the authors have verified that thiol is an intermediate in the reaction, and it can also directly react with relatively high yield, why not directly use thiols as a substrate for reaction screening? Thiol substrates are cheaper and more readily available, which would make the study more meaningful, and would be a more significant contribution to the construction of alkyl or arylboronic esters.*

Reply: We thank the reviewer for this comment and realize that the wording has been confusing. The benzylic thiol is an intermediate only when disulfides are used as starting materials. In the case of the S-aryl thioethers, a thiophenol equivalent is released but it does not undergo desulfurization. The text has been revised for clarity. The benefit of using thioethers instead of thiols is the higher functional group tolerance of the former, which is the result of their more anodic reduction potential that allows for the formation of the key carbanionic intermediate at milder conditions.

6. *Moreover, the paper offers little expansion on the thiols and disulfides substrates, and does not reflect the compatibility of these two types of substrates.*

Reply: We thank the reviewer for the input and have expanded the scope of thiols and disulfides. Successful examples are found in the updated Figure 1 in the manuscript, whereas unsuccessful substrates were added to Supplementary Information (Section S7). In addition to this expanded scope, successful use of thioacetals as starting materials for desulfurative borylation has been included in the revised Figure 1.

7. *In the radical trapping experiment, the lifetime of the primary free radical is short and is easily quenched, making it difficult to obtain cyclization products. More stable tertiary radicals should be considered for this experiment.*

Reply: We thank the reviewer for the input. Unfortunately, the starting material for the tertiary analogue of compound **1ak** is synthetically challenging to obtain and after several attempts, using a variety of methods, we decided to synthesize a thioether with a cyclopropyl group (**1al**) appended to the benzylic position instead. Due to the propensity of such cyclopropyl groups to undergo radical ring opening, electrolysis of this substrate would be complementary to the cyclization attempt using thioether **1ak**. However, electrolysis of **1al** under standard conditions resulted in clean formation of the corresponding borylated product **2ao** in 79% yield with the cyclopropyl ring intact, providing further support for the proposed anionic cross-coupling mechanism.

8. *The author cited literature [51] in the article, but it is not listed in the references.*

Reply: We thank the reviewer for this comment. The reference list has been updated in the revised manuscript.

Reviewer #3 (Remarks to the Author):

1. *The authors claimed that low-valent sulfur-containing compounds are abundant among natural products. However, most of substrates in Scheme 1 are prepared from benzyl bromides and thiophenols. Moreover, the electrochemical borylation of alkyl halides has been reported (J. Am. Chem. Soc. 2021, 143, 12985-12991). Based on these concerns, the reviewer could not support the publication of this manuscript in Nature Communications.*

Reply: We acknowledge the concerns raised by the Reviewer. However, while the thioethers used here were indeed synthesized from the corresponding benzylic and allylic bromides for simplicity, there is a plethora of synthetic methods available that enables access to this compound class from thiols. The revised manuscript has been updated to emphasize this with the addition of references 73-100. Furthermore, the revised Supplementary Information includes a demonstration of this approach for the preparation of the benchmark thioether **1a** from the corresponding thiol (see Supplementary Information, Section S5.1). On a more general note, the development of methods that enable the use of new types of starting materials for synthesis of a known product class is valuable for the community as it expands the synthetic scope. The use of thiols and their derivatives to furnish borylated products under electrochemical conditions has, to the best of our knowledge, not previously been reported. In addition to an expanded scope of these compound classes, the revised manuscript now includes examples of the successful conversion of thioacetals into monoborylated products – a compound class that, as far as we are aware, has previously not been used in borylation reactions under any set of synthetic conditions. Furthermore, the work by Qi, Lu and co-workers that the Reviewer refers to proceeds using benzyl bromide as substrate with a diborane as coupling partner in a radical mechanism. Notably, this mechanism is distinct to that of the herein reported transformation, which appears to proceed via carbanion attack onto HBPIn as a highly unconventional electrophile. For comparison, we have carried out a reaction that utilizes benzyl bromide instead of thioether **1a** under our benchmark conditions. In this case, product **2a** failed to form (see Supporting Information, Section S7), clearly indicating the distinct nature of this borylation procedure that offers a new route to benzylic and allylic pinacolboranes.

2. *Electrochemical borylation of alkyl halides and alkanes should be discussed in the background section. These are important methods for obtaining boronic esters.*

Reply: We thank the reviewer for this comment and have expanded the introduction for improved contextualization of this work.

3. *A lot of natural thioethers should be selected and used for the exploration of the substrate scope.*

Reply: We thank the reviewer for the input. The number of commercially available aryl benzyl thioethers of natural, as well as pharmaceutical, origin is limited and those tested were not amenable for borylation (see Supplementary Information, Section S7). However, successful borylation of thioethers and thioacetals derived from compounds such as rosuvastatin, salicin, fenofibrate, myrtenol, geraniol, roflumilast and terfenadine are found in the scope in Scheme 1 in the main text.

4. *Why MeCN instead of THF was used in the gram scale synthesis.*

Reply: We thank the reviewer for pointing out this typo, which is now removed from the updated manuscript and SI.

5. *Why primary and tertiary boronic esters (2a and 2s) were obtained in high yields but secondary boronic ester 2t was isolated in only 42% yield, even though 10 equiv. HBpin were used as reaction partner.*

Reply: We thank the Reviewer for this comment that prompted us to re-run a series of thioethers of the primary, secondary and tertiary in both MeCN and THF. Doing so, we found that reactions in THF typically result in higher yields and that the yields of primary, secondary and tertiary borylated products follow a trend that may be expected based on steric hindrance and Scheme 1 has been updated with these results in the revised manuscript. Nevertheless, a clear preference for borylation of the tertiary over the secondary thioether is still observed for the formation of compound **2y**, indicating that the interplay of sterics and electronics can result in substrate specific reactivity and selectivity patterns.

In this manuscript, Lundberg and coworkers have developed an electrochemical method for desulfurative borylation of thiols, disulfides and thioethers to synthesize a series of boronic esters. The authors proposed that the reaction was initiated by the cathodic reduction of thiols, disulfides or thioethers to give benzyl radicals, and then to carbanions. The mechanism was supported by cyclic voltammetry (CV) and radical trapping experiments.

The authors claimed that low-valent sulfur-containing compounds are abundant among natural products. However, most of substrates in Scheme 1 are prepared from benzyl bromides and thiophenols. Moreover, the electrochemical borylation of alkyl halides has been reported (J. Am. Chem. Soc. 2021, 143, 12985-12991). Based on these concerns, the reviewer could not support the publication of this manuscript in *Nature Communications*.

The reviewer would like to share the following comments and suggestions:

1. Electrochemical borylation of alkyl halides and alkanes should be discussed in the background section. These are important methods for obtaining boronic esters.
2. A lot of natural thioethers should be selected and used for the exploration of the substrate scope.
3. Why MeCN instead of THF was used in the gram scale synthesis.
4. Why primary and tertiary boronic esters (2a and 2s) were obtained in high yields but secondary boronic ester 2t was isolated in only 42% yield, even though 10 equiv. HBpin were used as reaction partner.